# Multimorbidity Profile of COVID-19 Deaths in Portugal during 2020

**DOI:** 10.3390/jcm11071898

**Published:** 2022-03-29

**Authors:** Paulo Jorge Nogueira, Miguel de Araújo Nobre, Cecília Elias, Rodrigo Feteira-Santos, António C.-V. Martinho, Catarina Camarinha, Leonor Bacelar-Nicolau, Andreia Silva Costa, Cristina Furtado, Liliane Morais, Juan Rachadell, Mário Pereira Pinto, Fausto Pinto, Antó Vaz Carneiro

**Affiliations:** 1Instituto de Medicina Preventiva e Saúde Pública, Faculdade de Medicina, Universidade de Lisboa, Avenida Professor Egas Moniz, 1649-028 Lisbon, Portugal; rodrigosantos@medicina.ulisboa.pt (R.F.-S.); lnicolau@medicina.ulisboa.pt (L.B.-N.); cristina.furtado@insa.min-saude.pt (C.F.); avc@medicina.ulisboa.pt (A.V.C.); 2Área Disciplinar Autónoma de Bioestatística (Laboratório de Biomatemática), Faculdade de Medicina, Universidade de Lisboa, Avenida Professor Egas Moniz, 1649-028 Lisbon, Portugal; 3Instituto de Saúde Ambiental, Faculdade de Medicina, Universidade de Lisboa, Avenida Professor Egas Moniz, 1649-028 Lisbon, Portugal; andreia.costa@esel.pt (A.S.C.); lilianemorais7@hotmail.com (L.M.); 4EPI Task-Force FMUL, Faculdade de Medicina, Universidade de Lisboa, Avenida Professor Egas Moniz, 1649-028 Lisbon, Portugal; mnobre@maloclinics.com (M.d.A.N.); ana.elias@arslvt.min-saude.pt (C.E.); ahmartinho@campus.ul.pt (A.C.-V.M.); ccamarinha@medicina.ulisboa.pt (C.C.); 5Clínica Universitária de Estomatologia, Faculdade de Medicina, Universidade de Lisboa, Avenida Professor Egas Moniz, 1649-028 Lisbon, Portugal; 6Unidade de Saúde Pública Francisco George, Agrupamento de Centros de Saúde Lisboa Norte, Administração Regional de Saúde de Lisboa e Vale do Tejo, 1500-559 Lisbon, Portugal; 7AIBILI—Associação para Investigação Biomédica em Luz e Imagem, Azinhaga Sta, Comba, Celas, 3000-548 Coimbra, Portugal; 8Unidade de Epidemiologia, Instituto de Medicina Preventiva e Saúde Pública, Faculdade de Medicina, Universidade de Lisboa, Avenida Professor Egas Moniz, 1649-028 Lisbon, Portugal; 9CIDNUR—Centro de Investigação, Inovação e Desenvolvimento em Enfermagem de Lisboa Escola Superior de Enfermagem de Lisboa, Avenida Professor Egas Moniz, 1600-190 Lisbon, Portugal; 10CRC-W—Católica Research Centre for Psychological, Family and Social Wellbeing, Universidade Católica Portuguesa, Palma de Cima, 1649-023 Lisbon, Portugal; 11National Institute of Health Dr. Ricardo Jorge, Av. Padre Cruz, 1600-560 Lisbon, Portugal; 12Unidade de Saúde Pública, Agrupamento de Centros de Saúde Lisboa Ocidental e Oeiras, Administração Regional de Saúde de Lisboa e Vale do Tejo, 2770-219 Lisbon, Portugal; juan.rachadell@campus.ul.pt; 13Casa Civil da Presidência da República, Presidência da República de Portugal, 1349-022 Lisbon, Portugal; mpinto@presidencia.pt; 14Centro Cardiovascular da Universidade de Lisboa, Centro Académico de Medicina de Lisboa, Faculdade de Medicina, Universidade de Lisboa, Avenida Professor Egas Moniz, 1649-028 Lisbon, Portugal; faustopinto@medicina.ulisboa.pt; 15Departamento do Coração e Vasos, Centro Hospitalar Universitário Lisboa Norte (CHULN), EPE, Avenida Professor Egas Moniz, 1649-028 Lisbon, Portugal; 16Instituto de Saúde Baseada na Evidência, Faculdade de Medicina, Universidade de Lisboa, Avenida Professor Egas Moniz, 1649-028 Lisbon, Portugal; 17Cochrane Portugal, Faculdade de Medicina, Universidade de Lisboa, Avenida Professor Egas Moniz, 1649-028 Lisbon, Portugal

**Keywords:** COVID-19, mortality, comorbidity, Charlson comorbidity index, Elixhauser comorbidity index, Portugal

## Abstract

Background: COVID-19 is caused by SARS-CoV-2 infection and has reached pandemic proportions. Since then, several clinical characteristics have been associated with poor outcomes. This study aimed to describe the morbidity profile of COVID-19 deaths in Portugal. Methods: A study was performed including deaths certificated in Portugal with “COVID-19” (ICD-10: U07.1 or U07.2) coded as the underlying cause of death from the National e-Death Certificates Information System between 16 March and 31 December 2020. Comorbidities were derived from ICD-10 codes using the Charlson and Elixhauser indexes. The resident Portuguese population estimates for 2020 were used. Results: The study included 6701 deaths (death rate: 65.1 deaths/100,000 inhabitants), predominantly males (72.1). The male-to-female mortality ratio was 1.1. The male-to-female mortality rate ratio was 1.2; however, within age groups, it varied 5.0–11.4-fold. COVID-19 deaths in Portugal during 2020 occurred mainly in individuals aged 80 years or older, predominantly in public healthcare institutions. Uncomplicated hypertension, uncomplicated diabetes mellitus, congestive heart failure, renal failure, cardiac arrhythmias, dementia, and cerebrovascular disease were observed among COVID-19 deceased patients, with prevalences higher than 10%. A high prevalence of zero morbidities was registered using both the Elixhauser and Charlson comorbidities lists (above 40.2%). Nevertheless, high multimorbidity was also identified at the time of COVID-19 death (about 36.5%). Higher multimorbidity levels were observed in men, increasing with age up to 80 years old. Zero-morbidity prevalence and high multimorbidity prevalences varied throughout the year 2020, seemingly more elevated in the mortality waves’ peaks, suggesting variation according to the degree of disease incidence at a given period. Conclusions: This study provides detailed sociodemographic and clinical information on all certificated deaths from COVID-19 in Portugal during 2020, showing complex and extreme levels of morbidity (zero-morbidity vs. high multimorbidity) dynamics during the first year of the pandemic in Portugal.

## 1. Introduction

The COVID-19 pandemic has plunged the world into an unprecedented health, economic, and social crisis [1]. As of 8 September 2021, the severe acute respiratory syndrome coronavirus 2 (SARS-CoV-2), which emerged in China in December 2019, has infected more than 222 million individuals worldwide and generated 4.5 million COVID-19-related deaths [1,2]. Complications such as acute respiratory distress syndrome (ARDS), acute kidney injury (AKI), myocardial injury, and death may occur [3,4].

The first COVID-19 patients were diagnosed in Portugal on 2 March 2020. Since then, more than one million cases and more than 17,000 deaths have been reported in Portugal. Indeed, the introduction of this new disease has been associated with an increase in the number of all-cause deaths compared with the average number of all-cause deaths in the previous five years. A total of 99,356 deaths were reported in Portugal between 2 March and 27 December, an increase of 12,852 deaths compared to the mean of the previous five years in the same period [5]. From these excess deaths, Statistics Portugal, I.P. (INE), reported that 6683 (52%) were directly due to COVID-19 [5].

Several individual [6] and clinical characteristics [7,8] have been associated with poor COVID-19 outcomes. Age has been the most consistently associated risk factor with death [6]. The male sex [9], race/ethnicity [10], and pre-existing comorbidities, such as cardiovascular disease, hypertension, diabetes, chronic respiratory diseases, cerebrovascular disease, malignancy, kidney disease, and liver disease [11], can also exert a significant impact on the prognosis of the disease [12].

Assessing the distribution and coexistence of morbidities and estimating their impact on death prognosis allow for the identification of high-risk COVID-19 patients. The Charlson et al. [12] and the Elixhauser et al. [13] indexes are commonly used to create comorbidity risk-adjustment models for individuals’ risk and outcome predictions. These indexes are available for coding 17 and 31 different medical conditions, respectively, according to the International Statistical Classification of Diseases and Related Health Problems, 10th Revision (ICD-10) [14]. Previous studies on severe and critical COVID-19 patients using these indexes have shown them to be able to stratify these patients’ risk of death adequately [15,16].

Our study aimed to characterize the morbidity profiles of fatalities due to COVID-19 infection in Portugal, intending to contribute evidence for potentially improving the evaluation of COVID-19 patients and optimizing medical interventions and approaches.

## 2. Materials and Methods

### 2.1. Study Population and Data Sources

This study used retrospective data of death certificates during 2020 in Portugal to characterize the distribution of comorbidities and characteristics of COVID-19-related deaths. The dataset used is a subset of the National e-Death Certificates Information System (Sistema de Informação do Certificados de Óbito) (SICO). SICO is the Portuguese national mortality information system. It is a web-based system used by Portuguese medical doctors to certify deaths and has been fully electronic since 1 January 2014. The Portuguese Directorate-General of Health (DGS)—the technical and normative arm of the Portuguese Ministry of Health—is responsible for managing the SICO database and provided the dataset used here. Medical doctors write causes of death and comorbidities in the database in an open text form. Subsequently, the DGS’s team of specialized coders for mortality coded the causes of death and related comorbidities.

For this study, all deaths certificated in Portugal (mainland and the Autonomous Regions of Azores and Madeira) in which the underlying cause of death was “COVID-19” (ICD–10 codes: U07.1 or U07.2) and registered between 16 March and 31 December 2020 were included in this study. The codes used to identify COVID-19-related deaths followed the coding guidelines of the WHO for the COVID-19 pandemic [17].

Resident population estimates for Portugal are available at the Statistics Portugal (INE) website [18] for 2020 to calculate global crude mortality rates and age-specific rates per 100,000 individuals. Population estimates were segregated according to territorial units for statistics classification, level 2 (NUTS II; approximately Health Administrative Regions (ARS)), districts, municipalities, and age groups.

Additional data characterizing the number of COVID-19 daily infections were obtained from the https://github.com/CSSEGISandData repository (accessed on 21 December 2021) [2].

### 2.2. Population Characteristics and Morbidity

Population characteristics available in the SICO dataset were described, such as age (recoded into age groups), sex, region, health region, district, municipality, and death place.

Comorbidities were derived from the ICD10 codes associated with each death certificate, which include the codification of the underlying cause of death, diseases, and other conditions. Comorbidities relating to each COVID-19 death were then used to calculate the Elixhauser index [13] as a primary approach and the Charlson index [12] as a supplementary approach. These indexes are methods of categorizing patient comorbidities according to respective ICD-10-adapted algorithms (Appendix B). Elixhauser-associated indexes were adopted for statistical analysis: the Elixhauser index, the weighted Elixhauser with Agency for Healthcare Research and Quality (AHRQ) modification [19], and the Elixhauser index with the van Walraven modification [20]. These weighted indexes stratify the respective weighted scores into four groups: <0, 0, 1–4, and ≥5 (i.e., from less probability of death, or potentially low morbidity, to a greater likelihood of death, or potentially high multimorbidity). The place-of-death variable was constructed from the available text descriptions in the database.

A variable dividing the pandemic into three periods during 2020 was also created and used to compare COVID-19-related deaths in each stage. Each period had distinct characteristics in the evolution of COVID-19 infections, according to the observed 7 days incidence and assuming that two infectious waves occurred, with period 1 as the first infection wave (between 2 March and 2 June); period 2 as a calmer infectious stage (between 3 June and 11 August); and period 3 as the initial increasing period of the extensive second wave of infections, starting on 12 August and lasting until 31 December (the second wave of infections went on at least until March 2021; however, it was mostly outside of the defined period for this study).

### 2.3. Statistical Analysis

Descriptive statistics of single variables were performed using absolute and relative frequencies, means, medians, standard deviations (SDs), interquartile ranges (IQRs), and minimum as well as maximum statistics.

Global crude mortality rates, age-specific rates, and age-standardized rates per 100,000 individuals were calculated using the absolute number of deaths divided by the estimated resident population’s respective number (obtained at the Statistics Portugal (INE) website). The WHO 2000–2025 standard population [21] was used for the calculation of age-standardized rates. Additionally, the absolute number of deaths and rate ratios per sex were calculated for the main variables and morbidities.

Comparisons of quantitative variables were done descriptively and using nonparametric tests (Mann–Whitney U tests for two groups and Kruskal–Wallis tests for three or more groups). When necessary, associations between categorical variables were performed using Fisher’s exact test resorting to Monte Carlo (simulated *p*-value). Distributions’ homogeneity by category variables within each comorbidity was tested using Pearson’s goodness-of-fit test, considering each categorical variable’s total/marginal distribution.

For all the tests performed, the level of statistical significance was set at 0.05. All the analyses were conducted using the R software version 3.6.3 (R Foundation for Statistical Computing, Vienna, Austria). The determination of comorbidities was performed using the R “Comorbidity” package, version 0.5.3 [22].

## 3. Results

### 3.1. Study Population Characteristics

During 2020, a total of 6701 COVID-19-related deaths were recorded in the e-death certificate database in Portugal: 3502 males and 3199 females, with an average age (SD) and median of 82.0 (10.6) and 84 years, respectively. Two COVID-19-related deaths peaks were observed in Portugal, the first during the 13th week and the second in the 50th week of 2020 (Appendix A). It is noteworthy that the highest number of deaths per week was similar between the sexes during the first peak and co-occurring. During the second peak, the highest number of deaths per week was lowest for men and occurred about two weeks earlier (Appendix A). Both peaks of deaths per week reflect different evolutions in the incidence of COVID-19 across the three observed pandemic stages: two waves of infections separated by a period with a consistent and low number of COVID-19 cases per week (Appendix A).

Regarding the distribution of deaths by age group, more than two-thirds of COVID-19-related deaths were in individuals aged 80 years and older. The distribution of deaths by sociodemographic variables and death rates are detailed in Table 1. The global death rate was estimated at 65.5 deaths per 100,000 inhabitants, higher in males (72.1) than females (58.8). Additionally, the death rate per 100,000 inhabitants was differently distributed by region. In more densely populated regions with over 1,000,000 inhabitants (North, Center, and Lisbon Metropolitan Area), higher crude mortality rates (CMRs) were registered in men; however, in less densely populated regions with fewer than 1,000,000 inhabitants (Alentejo, Algarve, and both Autonomous Regions of the Azores and Madeira), higher CMRs were registered in women.

It is noteworthy that although, globally, the ratio between male and female COVID-19 deaths was estimated as 1.1 (approximately 10% more men than women), within age groups, the panorama was substantially different. From 20 to 70 years, absolute deaths were consistently higher in males within each age decade. Most total death number ratios by age decade were more than 1.5-fold higher in males (namely, in individuals in their 20s, 30s, and 70s) and about 2-fold in those in their 40s and 50s, while a 2.7-fold-higher difference was verified in those aged in their 60s. The mean age of death in men and women was statistically different (*p* < 0.001), with 84.3 years in females and 79.9 years in males (Appendix A). In relative terms, using CMR ratios, this discrepancy was even more remarkable, with 5- to almost 12-fold male-to-female mortality difference within age groups (Table 1). The ratio between both sexes’ age-standardized mortality rates consistently indicated a global 6.9 for males compared to females. In terms of geographical distribution, the highest absolute number of deaths was observed in Porto at the district level, and the highest mortality rate was in Bragança. At the municipal level, the highest absolute number of fatalities and the highest mortality rate were observed in Lisbon and the municipality of Vimioso, respectively (Appendix A). The Northern ARS registered the most elevated mortality at the regional level in the absolute number of COVID-19 deaths and mortality rates (Appendix A). The most COVID-19-related deaths occurred mainly in a “public health institution” (90.8%), with a higher proportion in males (93.3%) than in females (88.0%). Considerably fewer, 4.3%, deaths occurred “at home” (slightly more in females than males). Moreover, the third-highest place of death was identified as at a “nursing home” with 4.2%, this proportion being 2.2-fold higher in females than in males (Table 1). A very residual number of deaths was identified as occurring in a “private health institution” (just 0.2%).

Considering the age distribution by death location, a higher mean age at death was observed in the “unknown” and “nursing homes” categories (84.4 and 87.1 years, respectively). In comparison, lower mean ages were observed for deaths occurring in “public health institutions” and “at home” (81.8 and 82.6 years, respectively) (Appendix A).

### 3.2. Comorbidities

Among the deceased, a total of 29,775 ICD-10 codes were registered, with an average of 4.44 codes per death (SD: 3.04; range: 1 to 15). Evaluating this information according to the presence of comorbidities on Elixhauser’s list, a total of 9234 occurrences were observed. The number of comorbidities per death ranged from 0 to 8, with an average (SD) of 1.38 (1.44) and a median of 1. Thus, about 37.9% of deaths had zero Elixhauser’s comorbidities associated with them, while 40.9% had at least two Elixhauser’s comorbidities identified with this approach. The top ten Elixhauser’s comorbidities associated with COVID-19 fatalities that occurred in Portugal during 2020 included (in order of importance): uncomplicated hypertension (30.0%), uncomplicated diabetes mellitus (16.8%), congestive heart failure (14.5%), renal failure (12.8%), cardiac arrhythmias (10.6%), pulmonary circulation disorders (7.1%), solid tumors (7.1%), obesity (5.2%), other neurological disorders (4.8%), and fluid as well as electrolyte disorders (3.8%) (Table 2 or Appendix A for a complete table).

The presence of comorbidities differed between the sexes. For example, in Elixhauser’s comorbidities, associated morbidity rankings differed slightly, as renal failure was third in women and fourth in men. In addition, several morbidities were shown to differ by sex, such as chronic heart failure, solid tumors, obesity, hypothyroidism, depression, and rheumatoid arthritis/collagen vascular disease. At the same time, age groups were significantly associated with almost all morbidities.

Additionally, there were no substantial differences by health region of residence, where only two morbidities showed some heterogeneity: depression and blood loss anemia. Additionally, comparing the three defined periods of the pandemic in Portugal, it is noteworthy that the morbidity rankings vary from period to period. However, no substantial heterogeneities were identified within morbidity (exceptions were fluid and electrolyte disorders; weight loss; and AIDS or HIV) (Appendix A).

The construction of Charlson’s comorbidities generated 6605 occurrences that are displayed in Appendix A. The number of comorbidities per death ranged from 0 to 6, with an average (SD) of 0.99 (1.03) and a median of 1. Thus, approximately 40.3% of deaths had zero Charlson comorbidities associated with them. On the other hand, 28.2% of deaths had at least two Charlson comorbidities identified. It is noteworthy that two comorbidities are identical in the Charlson and Elixhauser indexes: congestive heart failure (CHF) and peripheral vascular disorders (PVD), with CHF ranking in the top three in both approaches. The top five Charlson comorbidities included uncomplicated dementia, diabetes, CHF, renal disease, and cerebrovascular disease. The inclusion of dementia in the Charlson top five is an important departure from the Elixhauser approach, but overall, the top morbidities in both approaches show a similar message.

Additionally, no substantial differences by sex were observed (with the exceptions of dementia, chronic heart failure, malignant cancer, and rheumatoid disease). Here, all morbidities showed heterogeneity by age groups.

In COVID-19 deaths observed in Portugal during 2020, correlations between comorbidities observed together and identified using Elixhauser were few. Although no remarkable negative correlations were found, some positive correlations were observed. The correlation between pairs of comorbidities seems to be even less relevant in what concerns Charlson’s estimated comorbidities. However, a slight negative correlation was observed between dementia and several remaining comorbidities (Figure 1B).

Elixhauser’s list: chf—congestive heart failure; carit—cardiac arrhythmias; valv—valvular disease; pcd—pulmonary circulation disorders; pvd—peripheral vascular disorders; hypunc—hypertension, uncomplicated; hypc—hypertension, complicated; para—paralysis; ond—other neurological disorders; cpd—chronic pulmonary disease; diabunc—diabetes, uncomplicated; diabc—diabetes, complicated; hypothy—hypothyroidism; rf—renal failure; ld—liver disease; pud—peptic ulcer disease, excluding bleeding; aids—AIDS/HIV; lymph—lymphoma; metacanc—metastatic cancer; solidtum—solid tumor, without metastasis; rheumd—rheumatoid arthritis/collaged vascular disease; coag—coagulopathy; obes—obesity; wloss—weight loss; fed—fluid and electrolyte disorders; blane—blood loss anemia; dane—deficiency anemia; alcohol—alcohol abuse; drug—drug abuse; psycho—psychoses; and depre—depression.

Charlsons’ list: ami—acute myocardial infarction; chf—congestive heart failure; pvd—peripheral vascular disease; cevd—cerebrovascular disease; dementia—dementia; copd—chronic obstructive pulmonary disease; rheumd—rheumatoid disease; pud—peptic ulcer disease; mld—mild liver disease; diabetes—diabetes without complications; diabetes w/com—diabetes with complications; hp—hemiplegia or paraplegia; rend—renal disease; cancer—cancer (any malignancy); msld—moderate or severe liver disease; metacanc—metastatic solid tumor; and aids—AIDS/HIV.

### 3.3. Multimorbidity

The distribution of COVID-19 deaths according to Elixhauser morbidity composite scores revealed high percentages of deceased individuals associated with low morbidity (low expected probability of death) and with high multimorbidity (high expected probability of death) (Table 3). Higher levels of multimorbidity (present in COVID-19 fatalities) were more frequently estimated in men than in women. By age group, the morbidity indexes’ analyses are more challenging. Overall, morbidity increases with age. However, in the 80 years old and older group, a slight reduction in multimorbidity and a concomitant increase in the percentual occurrence of death were observed to be associated with nonidentified morbidity. The distribution patterns of fatalities along the various comorbidity categories of both the Elixhauser and Charlson indexes varied between residence districts, suggesting differences between districts with higher (Bragança and Porto) and lower (Beja and Faro) mortality rates. The same is observed at the level of health regions. More than 90% of deaths occurred in a “public health institution”, influencing the overall observed morbidity profile. Deaths observed “at home” seem to have happened in individuals with lower levels of morbidity.

In contrast, deaths registered as occurring in “nursing homes” revealed a morbidity profile similar to those found globally and in health institutions, with slightly lower multimorbidity. The number of deaths reported in “private health institutions” was very low, accounting only for 0.2% of deaths. Considering the Elixhauser composite scores as well as the three periods of 2020 defining the evolution of the pandemic, evidence emerged that the morbidity profile of observed deaths differed in each period. The differences suggest higher multimorbidity levels during lower infection incidence phases and an association between higher infection incidence stages and somewhat lower multimorbidity profiles.

To further investigate the notion that the consecutive periods had different morbidity profiles, the Elixhauser indexes were separately analyzed for each period. A comparison between levels of multiple Elixhauser morbidities observed per death in the three pandemic periods is detailed in Appendix A. Considering the occurrence of two or more morbidities (Elixhauser score > 1), the percentage of individuals with this level of multimorbidity was 38.1% in the first period (first infection wave), 42% between the first and second infection waves, and 26.6% in the third period (ascending phase of the second infection wave). When considering the simultaneous presence of three or more morbidities (Elixhauser score > 2 and Elixhauser score > 3), the pattern was similar, although the difference between periods one and two was minimal. Still, in Appendix A, the extreme level of Elixhauser morbidity conveys a similar message. However, the score’s weighted versions that account more closely for the probability of death at hospital admission, considering the respective morbidities, show a more evident discrepancy among the three periods. In fact, extreme multimorbidity is higher during period two (lower incidence of the disease), while period one seems to have about 5% less excessive multimorbidity in its observed COVID-19 deaths. Last, the morbidity observed in period three has nearly half of the previous period’s extreme level of morbidity.

## 4. Discussion

This study assessed a Portuguese national death certificate database and retrospectively included all 6701 individuals deceased from COVID-19 between 16 March and 31 December 2020. The authors used comorbidities registered as directly contributing to death. The current methodology represents a strength of the present study, enabling the characterization of underlying comorbidities of COVID-19-related deaths.

Registered increased mortality for men was observed, characterized by an overall modest absolute increase but a dramatic increase within the age groups when applying subgroup analysis. Besides the sex differences in biological terms [23], the possible explanations for this result include the following: (a) the fact that women are more likely to search for healthcare services than men [24], accessing medical care at an earlier stage of COVID-19 disease; (b) the generally stronger response to infectious pathogens in women [25]; (c) the increased likelihood of the adoption of nonpharmaceutical measures to avoid infection, particularly hand hygiene [26] and preventive behaviors [27]; or (d) sex-based differences in multimorbidity registered in this study that may be associated with severe states of COVID-19 [19,20,28,29,30,31,32,33].

The Elixhauser comorbidity index was not used to predict mortality in this study, given that we did not have enough data to do so. However, given the statistical superiority of the Elixhauser comorbidity system in predicting mortality outcomes [32] and the present study’s large dataset (with a reduced risk of overfitting), this index was used as a primary multimorbidity analysis tool, to the detriment of the Charlson comorbidity index, and revealed to be advantageous. The authors used two composite score modifications [19,33] to discriminate associations between multimorbidity and mortality by COVID-19. This allowed for managing the computational challenge in the current dataset using the Elixhauser comorbidity index. This strategy was previously reported with high accuracy in the literature for studying in-hospital mortality [20].

The data show that 37.9% (*n* = 2541) of COVID-19 deaths did not have Elixhauser morbidities, potentially indicating a high proportion of fatalities without comorbidities. About 11.2% (*n* = 284) and 3.7% (*n* = 95) of the deceased patients without death codes related to the Elixhauser list of comorbidities were under 70 and 60 years of age, respectively. The demographic analysis of this group of patients revealed a higher number of men (*n* = 1338) compared to women (*n* = 1203) and a mean age of 82.5 years, with 69.9% (*n* = 1771) of the patients aged 80 years or more. This result hints at a high global absence of morbidities, which is highly unlikely considering the Portuguese senior population’s characteristics. This finding may be related to an issue with death records, since these patients died in periods of an increased number of deaths, which could have led to less time to certify each death and thus less time to register comorbidities. The absence of comorbidities assumed by the Elixhauser index can also explain this issue. Of the 2541 patients, 306 (12.0%) had dementia, considering the Charlson comorbidity list. According to a systematic review, this condition is one of the identified prognostic factor pathologies for mortality by COVID-19, with a 54% increase in the odds [34].

The data suggested that the presence of multimorbidity was strongly concomitant with fatalities by COVID-19, as multimorbidity was registered in 40.9% of deaths. Previous studies reported the association of several comorbidities with an increased mortality rate. Using surveillance data of the first 20,293 cases during the initial stage of the pandemic in Portugal, a previous study investigated the role of preconditions in COVID-19 deaths. In that study, cardiac diseases, kidney disorders, and neuromuscular disorders were registered as potential risk indicators for COVID-19 deaths [35]. The same study also registered a detrimental effect on COVID-19 deaths for several comorbidities. For example, cardiac diseases and renal failure—included in the top 10 diseases present at the time of death—cancer, hypertension (uncomplicated), diabetes (uncomplicated), congestive heart failure, obesity, and chronic pulmonary disease were also indicated as prognostic factors for severity or mortality in patients infected with COVID-19. These associations were also recognized in systematic reviews and meta-analyses on the subject [34,36,37,38,39,40,41,42].

A COVID-19 pandemic early-stage study reported estimating excess mortality due to COVID-19, nonidentified COVID-19, and a decrease in healthcare access of approximately 2400 to 4000 deaths [43]. This three-partied mechanism could partly explain the high lethality in multimorbidity patients registered in this analysis. Furthermore, a systematic review reported an association between multimorbidity and increased loneliness levels [44], implying decreased healthcare access for individuals highly susceptible to severe COVID-19 outcomes. However, the number of studies investigating the effect of multimorbidity on the lethality of COVID-19 is limited and in great need of expansion. A similar association between multimorbidity and COVID-19 lethality was registered in a study investigating populations from the South American continent. That study reported different diseases, including high blood pressure, diabetes mellitus, obesity, and cardiovascular, respiratory, and chronic kidney disease [45]. Therefore, the authors consider it very important that the results of the present study using the current methodology confirmed the association of multimorbidity with COVID-19 death suggested by previous studies in the literature [34,35,36,37,38], providing a point of reference for future research.

Considering the geographical location, the epidemiological assessment of COVID-19 mortality is complex, given the distinct perspectives of mortality absolute figures and rates, especially in less densely populated areas. Moreover, considering the region or district of residence as a factor for COVID-19 mortality lacks direct epidemiological plausibility. It implies a biased causal path given its association with latent unobserved factors, such as socioeconomic factors, the timing of the introduction and propagation of the disease, public health metrics (e.g., case detection and testing capacity, healthcare service practices and proficiency, and the implementation of mitigation strategies), and its condition as the ancestor of exposure and outcomes with both age and preconditions [46,47,48]. The authors discuss the region/district of residence only in the COVID-19 geographical fatality distribution context. When evaluating the health regions and districts of residence with at least 20 deaths, the cutoff for a higher fatality ratio in men was areas with over 1,000,000 inhabitants (districts located in the North, LVT, and Center administrative health regions). In comparison, women’s fatality ratio was higher in less densely populated areas in southern Portugal (below 1,000,000 inhabitants). In these less densely populated regions and districts, the average age of death was above 80 years. This result was expectable in terms of observing a higher death ratio for women given the significant difference between the sexes in the average age of death (79.8 years for men; 84.3 years for women). The exception was the Guarda district, where the CMR ratio was higher for women. Still, this district’s average age of death was also the highest in Portugal, above two standard deviations from the national average.

The present study’s mortality data suggested a difference in the characteristics of the affected population between the three time periods. A decrease in mortality was observed for individuals with higher multimorbidity levels, while an increase in mortality was observed for individuals with lower multimorbidity levels. Several potential reasons arise for the explanation of this result. First, the completeness of the records decreased in period three (average of 1.39 morbidities per individual (in 5046 deaths) in period three compared to an average of 1.46 morbidities per individual (in 264 deaths) in period two and an average of 1.29 morbidities per individual (in 1332 deaths) in period one). Second, the significant increase in mortality that occurred in time period three might have caused the collapse of health services. Last, the Alpha variant of the SARS-CoV-2 virus, which may be responsible for higher mortality [49], became predominant in Portugal in period three. This result warrants further research to clarify the health system or etiopathogenesis mechanisms involved in these results. Administrative datasets help study and estimate health outcomes [28]. Still, the data are more vulnerable to potential sources of bias and limitations, including inaccuracy, missing data, or a lack of updating. Using administrative datasets, measuring multimorbidity (as the existence of multiple conditions ≥ 2 in the same person) [29] can be complex. To overcome administrative data limitations, the use of proper comorbidity adjustment is essential [30,31].

Despite being positively impressed with the comprehensiveness and detail of causes of death codes, the authors acknowledge the effect of death certificate errors as a limitation to the present study. They may also lead to a potential underestimation of some causes, especially in the elderly population [50]. In health, information and communication technologies have manifested themselves as essential tools for providing health services. The e-death certification system (SICO platform) intended, from its conception, to optimize mortality data quality and epidemiological surveillance. It proposed to do so through the electronic certification of deaths, improving both the access to and quality of mortality data [51,52]. Moreover, the electronic certification of deaths allows electronic communication between information systems with relevant information associated with cause of death and worldwide comparability (ICD-10 coding) [53,54]. Standardized death certification and coding practices also ensure quality mortality data and accurate tracking of multimorbidity related to COVID-19 mortality, which are crucial in public health/epidemiological matters. However, data completeness may be somewhat compromised for several reasons, including their use by different physicians, the little clinical information available, the absence of close contacts, and difficulty prioritizing the conditions leading to death in elderly patients due to multimorbidity [50,55]. Additionally, the highly stressful work environment due to the current pandemic crisis may decrease the records’ quality. There is also the possibility that the high level of detail observed may stem from the particular focus on COVID-19, potentiated by multiple sources of information circulating at health authorities’ institutions, which may not be the case in normal conditions for all kinds of mortality. In fact, in regular conditions, year N codes of mortality are only available after October of year N + 1; hence, authors were working on these codes for COVID-19 deaths more than six months before that schedule.

The Elixhauser comorbidity index was chosen as the primary multimorbidity analysis tool, to the detriment of Charlson’s. The use of 30 disease indicators, statistical superiority in predicting mortality outcomes [32], and the present study’s large dataset (with reduced risk of overfitting) revealed advantages. To manage the computational challenge in the current dataset using the Elixhauser comorbidity index, the authors used two composite scores modifications [19,33] to discriminate associations between multimorbidity and mortality by COVID-19. This strategy was previously reported with high accuracy in the literature for studying in-hospital mortality [20]. Both the Elixhauser and Charlson indexes were essentially used here to identify diseases/comorbidities associated with COVID-19 deaths. As they are usually seen as in-hospital mortality predictive tools, and as about 90% of COVID-19 deaths were observed as occurring in a health institution, this might be considered a limitation in our study. Nevertheless, this study was not focused on in-hospital mortality predictions.

The present study detailed the characteristics of COVID-19 deaths in Portugal during the year 2020. Social demographic characteristics and underlying health conditions (morbidities) provide important clues that may be used to identify noninfected individuals with high risk (ages and multimorbidity patterns). Furthermore, targeting specific preventive measures/management and hospital triage can help maximize the probability of COVID-19 survival. Future research should include a qualitative assessment of the procedure used herein to obtain the comorbidities associated with COVID-19 deaths.

## 5. Conclusions

In the first year of the COVID-19 pandemic, considering the 6701 COVID-19 deaths in Portugal during 2020, the respective global mortality rate was estimated as being 65.5 per 100,000 inhabitants, with significantly different and higher mortality rates in men than in women. Overall, the mean age at death was 81.9 years old (higher within females), with the majority of these deaths occurring in individuals over 80 years old. However, differences between the male-to-female number of death ratios were observed within age groups, as this indicator ranged between 5- and 11-fold. Consistently, the ratio of age-standardized rates between men and women was 6.9-fold. In more densely populated regions, increased mortality was observed, and men predominated; in contrast, less-populated regions registered less mortality and a higher female proportion.

A high prevalence of zero morbidities was registered using both the Elixhauser and Charlson comorbidities lists, namely 53.8% in the VW Elixhauser index and 40.2% in Charlson’s. Nevertheless, the opposite is also true. In fact, a wide range of diseases/morbidities were identified at the time of COVID-19 death. The most prevalent were uncomplicated hypertension, dementia, uncomplicated diabetes mellitus, congestive heart failure, renal failure, and cardiac arrhythmias. Higher multimorbidity levels were observed in men, increasing with age up to 80 years old. The zero-morbidity prevalence and multimorbidity prevalence varied throughout the year 2020, seemingly more elevated in the mortality wave’s peaks, which could suggest a variation according to the degree of incidence at a given period.

This study provides detailed sociodemographic and clinical information on all certificated deaths with COVID-19 in Portugal during 2020, showing complex and extreme levels of morbidity (no morbidity vs. high levels of comorbidity) dynamics during the first year of the pandemic in Portugal.

## Figures and Tables

**Figure 1 jcm-11-01898-f001:**
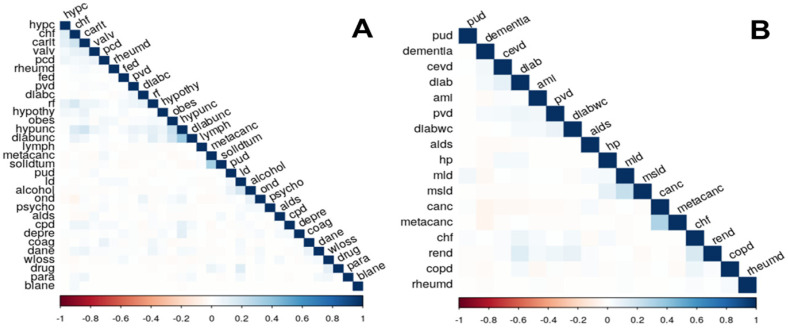
Pairwise correlations between morbidities of the Elixhauser comorbidity index (**A**) and the Charlson comorbidity index (**B**).

**Table 1 jcm-11-01898-t001:** COVID-19 death distributions and mortality rates (/100,000 individuals) by age group, district of residence, health region of residence, and location of death (total and according to sex).

	Total			Sex						Ratio Rates M/F	Ratio Deaths M/F	*p* ^1^
				Female			Male					
	*n*	%	Rate /10^5^	*n*	%	Rate /10^5^	*n*	%	Rate /10^5^			
Total	6701	100.0		3199	47.5		3502	52.5			1.1	0.001
Age Group												<0.001
0 to 9 years	1	0.0	0.1	1	0.0	0.0	0	-	0.0	0.0	-	
10 to 19 years	2	0.0	0.2	1	0.0	0.0	1	0.0	0.2	5.0	1.0	
20 to 29 years	5	0.1	0.4	2	0.1	0.1	3	0.1	0.5	6.6	1.5	
30 to 39 years	13	0.2	1.1	5	0.2	0.2	8	0.2	1.3	7.9	1.6	
40 to 49 years	60	0.9	3.8	21	0.7	0.5	39	1.1	5.2	11.3	1.9	
50 to 59 years	165	2.5	11.1	53	1.7	1.4	112	3.2	16.0	11.4	2.1	
60 to 69 years	537	8.0	40.9	146	4.6	5.8	391	11.2	64.6	11.2	2.7	
70 to 79 years	1372	20.5	137.2	515	16.1	24.2	857	24.5	197.3	8.1	1.7	
>80 years	4532	67.6	664.9	2448	76.5	152.4	2084	59.5	849.2	5.6	0.9	
Unknown	1	0.0	0.1	1	0.0	0.0	0	-	0.0	0.0	-	
Crude Mortality Rate			65.5			58.8			72.1	1.2		
Age-Standardized Mortality Rate			20.2			4.0			27.7	6.9		
District of Residence												0.094
Aveiro	439	6.6	64.6	193	6.0	54.0	246	7.0	68.8	1.3	1.3	
Beja	60	0.9	42.9	34	1.1	47.2	26	0.7	36.1	0.8	0.8	
Braga	652	9.7	78.9	329	10.3	76.0	323	9.2	74.6	1.0	1.0	
Bragança	153	2.3	124.4	70	2.2	107.9	83	2.4	127.9	1.2	1.2	
Castelo Branco	122	1.8	68.7	63	2.0	67.4	59	1.7	63.1	0.9	0.9	
Coimbra	212	3.2	52.1	100	3.1	46.4	112	3.2	52.0	1.1	1.1	
Évora	87	1.3	57.6	50	1.6	63.2	37	1.1	46.8	0.7	0.7	
Faro	59	0.9	13.5	30	0.9	13.0	29	0.8	12.6	1.0	1.0	
Guarda	123	1.8	86.1	70	2.2	92.3	53	1.5	69.9	0.8	0.8	
Leiria	222	3.3	48.4	109	3.4	45.3	113	3.2	47.0	1.0	1.0	
Lisboa	1451	21.7	63.1	674	21.1	55.1	777	22.2	63.6	1.2	1.2	
Portalegre	66	1.0	64.0	35	1.1	64.4	31	0.9	57.0	0.9	0.9	
Porto	1659	24.8	93.1	768	24.0	81.4	891	25.4	94.5	1.2	1.2	
Santarém	250	3.7	58.2	120	3.8	53.0	130	3.7	57.5	1.1	1.1	
Setúbal	480	7.2	56.3	230	7.2	51.2	250	7.1	55.6	1.1	1.1	
Viana do Castelo	130	1.9	56.8	65	2.0	52.8	65	1.9	52.8	1.0	1.0	
Vila Real	156	2.3	82.0	72	2.3	71.4	84	2.4	83.3	1.2	1.2	
Viseu	205	3.1	57.9	97	3.0	52.0	108	3.1	57.8	1.1	1.1	
Madeira Island	13	0.2	5.2	9	0.3	6.8	4	0.1	3.0	0.4	0.4	
Santa Maria Island	17	0.3	12.4	14	0.4	19.8	3	0.1	112.7	5.7	0.2	
Terceira Island	2	0.0	3.6	1	0.0	3.5	1	0.0	33.7	9.6	1.0	
Graciosa Island	1	0.0	23.9	0	-	0.0	1	0.0	1.4	-	-	
Unknown	142	2.1	-	66	2.1	-	76	2.2	-	-	1.2	
Health Region of Residence												0.013
ARS Alentejo	228	3.4	32.6	130	4.1	35.7	98	2.8	29.2	0.8	0.8	
ARS Algarve	59	0.9	24.4	30	0.9	24.0	29	0.8	24.8	1.0	1.0	
ARS Centro	964	14.4	43.2	468	14.6	39.9	496	14.2	47.0	1.2	1.1	
ARS LVT	2236	33.4	77.9	1049	32.8	68.7	1187	33.9	88.5	1.3	1.1	
ARS Norte	3039	45.4	85.2	1432	44.8	76.0	1607	45.9	95.5	1.3	1.1	
AR Açores	20	0.3	7.9	15	0.5	11.1	5	0.1	4.2	0.4	0.3	
AR Madeira	13	0.2	3.0	9	0.3	3.9	4	0.1	1.9	0.5	0.4	
Unknown	142	2.1	-	66	2.1	-	76	2.2	-	-	1.2	
Location of Death												<0.001
Unknown	41	0.6	-	27	0.8	-	14	0.4	-	-	0.5	
In a public health institution	6083	90.8	-	2815	88.0	-	3268	93.3	-	-	1.2	
At home	286	4.3	-	159	5.0	-	127	3.6	-	-	0.8	
Nursing home	280	4.2		193	6.0		87	2.5			0.5	
Private health institution	11	0.2	-	5	0.2	-	6	0.2	-	-	1.2	

Notes: ^1^ Homogeneity test. Abbreviations: ARS, administrative region of health; AR, autonomous region.

**Table 2 jcm-11-01898-t002:** Comorbidities present (identified according to the Elixhauser comorbidity index), total, by sex and age group.

Conditions	Total	Male	Female	*p* ^1^	<40 Years	40–49 Years	50–59 Years	60–69 Years	70–79 Years	80+ Years	*p* ^1^
Hypertension, uncomplicated	30.00	31.01	29.07	0.147	4.76	13.33	18.79	28.86	31.85	30.32	**<0.001**
Diabetes, uncomplicated	16.83	17.35	16.36	0.325	4.76	15.00	19.39	20.11	22.23	14.81	**<0.001**
Congestive heart failure	14.48	16.85	12.31	**<0.001**	9.52	3.33	9.70	9.68	12.83	15.91	**<0.001**
Renal failure	12.83	12.25	13.36	0.205	14.29	5.00	12.73	10.61	13.19	13.11	**<0.001**
Cardiac arrhythmias	10.63	10.57	10.68	0.886	4.76	3.33	7.88	7.45	9.69	11.52	**<0.001**
Chronic pulmonary disease	7.13	6.75	7.48	0.264	0.00	5.00	10.91	8.94	7.65	6.69	**<0.001**
Solid tumor, without metastasis	7.06	5.97	8.05	**0.001**	4.76	13.33	13.33	8.57	8.09	6.27	**<0.001**
Obesity	5.24	5.85	4.68	**0.038**	4.76	8.33	7.88	10.61	6.20	4.19	**<0.001**
Other neurological disorders	4.86	4.41	5.28	0.105	0.00	11.67	5.45	7.08	4.66	4.55	**<0.001**
Fluid and electrolyte disorders	3.84	4.00	3.68	0.507	0.00	5.00	2.42	3.54	3.28	4.10	**<0.001**
Blood loss anemia	0.03	0.03	0.03	0.949	0.00	0.00	0.00	0.00	0.00	0.04	0.323

Notes: ^1^ Homogeneity test; significant results are presented in bold font. Only the 10 most frequent conditions are presented; the full table is presented in the Appendix A.

**Table 3 jcm-11-01898-t003:** COVID-19 deaths distributions (as a percentage of total deaths) and the Charlson and Elixhauser comorbidity (AHRQ and VW) weighted indexes by sex, age group, health region of residence, district of residence, location of death, and pandemic period.

	Charlson Weighted Index					AHRQ Elixhauser Comorbidity Index					VW Elixhauser Comorbidity Index				
	0	1–2	3–4	≥5	*p*	<0	0	1–4	≥5	*p*	<0	0	1–4	≥5	*p*
Total	40.2	42.0	13.6	4.1		15.4	42.9	7.4	34.2		4.1	49.7	9.5	36.7	
Sex															
Female	40.1	42.3	13.9	3.8	0.599	15.9	42.7	6.9	34.5	0.277	4.8	49.5	8.8	37	**0.011**
Male	40.5	41.8	13.4	4.4		14.9	43.1	7.9	34		3.5	49.9	10.2	36.5	
Age Group															
0 to 18 years	100	0	0	0	**<0.001**	0	100	0	0	0.103	0	100	0	0	**<0.001**
19 to 49 years	50	31.2	10	8.8		10	50	5	35		6.2	50	11.2	32.5	
50 to 59 years	39.4	40	13.9	6.7		13.3	38.8	9.7	38.2		4.8	43	13.3	38.8	
60 to 69 years	40	40.6	13.2	6.1		16.4	39.7	9.5	34.5		7.1	44.9	13.2	34.8	
70 to 79 years	37.7	43.1	13.5	5.8		16.8	41.5	8.5	33.2		4.2	49.3	11.7	34.8	
80 to 106 years	40.9	42.2	13.8	3.2		15	43.8	6.8	34.4		3.7	50.5	8.2	37.6	
Health Region of Residence															
ARS Alentejo	39	43.4	14	3.5	**0.028**	19.7	38.6	7.5	34.2	**<0.001**	7	47.4	7	38.6	**<0.001**
ARS Algarve	30.5	44.1	16.9	8.5		20.3	35.6	3.4	40.7		3.4	47.5	10.2	39	
ARS Centro	39.7	43.4	13.2	3.7		14.6	43.4	7.5	34.5		2.2	50.4	8.4	39	
ARS LVT	37.2	42.4	15.1	5.3		15.8	38.2	7.7	38.2		3.9	45.8	10.4	39.9	
ARS Norte	42.6	41.4	12.7	3.3		14.9	46.4	7.4	31.3		4.7	52.2	9.4	33.7	
AR Açores	30	35	25	10		25	30	5	40		0	50	10	40	
AR Madeira	38.5	46.2	0	15.4		7.7	53.8	0	38.5		0	38.5	30.8	30.8	
District of Residence															
Aveiro	45.8	39.2	10.9	4.1	**<0.001**	12.8	51.5	6.6	29.2	**<0.001**	2.5	57.4	9.6	30.5	**<0.001**
Beja	40	48.3	11.7	0		26.7	30	11.7	31.7		11.7	40	11.7	36.7	
Braga	39.3	40.8	15.6	4.3		16	42.2	8.6	33.3		4.8	48.6	10.1	36.5	
Bragança	34	45.8	15	5.2		12.4	37.9	6.5	43.1		4.6	41.2	6.5	47.7	
Castelo Branco	41	47.5	9.8	1.6		18	45.1	6.6	30.3		4.9	54.9	5.7	34.4	
Coimbra	38.7	44.8	12.3	4.2		9.9	42.5	8	39.6		1.9	45.3	10.4	42.5	
Évora	43.7	41.4	12.6	2.3		17.2	43.7	5.7	33.3		4.6	55.2	4.6	35.6	
Faro	30.5	44.1	16.9	8.5		20.3	35.6	3.4	40.7		3.4	47.5	10.2	39	
Guarda	40.7	39.8	17.1	2.4		15.4	41.5	6.5	36.6		1.6	46.3	5.7	46.3	
Leiria	38.7	41.9	13.5	5.9		17.1	41	4.5	37.4		1.8	50	8.1	40.1	
Lisboa	36	42.5	15.4	6.1		15.4	37.6	6.9	40		3.9	45.3	10.3	40.5	
Portalegre	39.4	34.8	16.7	9.1		16.7	43.9	4.5	34.8		7.6	47	6.1	39.4	
Porto	44.1	41.7	11.8	2.5		15.7	46.8	7.9	29.5		5.2	53	10.4	31.3	
Santarém	39.2	39.6	16	5.2		14.8	40.4	6.8	38		3.2	46.8	8.8	41.2	
Setúbal	38.1	45.2	14.2	2.5		17.7	37.3	11.5	33.5		4.2	45.8	11.9	38.1	
Viana do Castelo	43.8	41.5	10.8	3.8		12.3	54.6	4.6	28.5		3.1	60.8	5.4	30.8	
Vila Real	37.8	42.9	16	3.2		13.5	47.4	3.8	35.3		4.5	50.6	3.2	41.7	
Viseu	40.5	43.9	12.7	2.9		14.1	43.9	9.8	32.2		1.5	50.7	7.3	40.5	
Madeira Island	38.5	46.2	0	15.4		7.7	53.8	0	38.5		0	38.5	30.8	30.8	
Santa Maria Island	29.4	29.4	29.4	11.8		23.5	23.5	5.9	47.1		0	41.2	11.8	47.1	
Terceira Island	50	50	0	0		50	50	0	0		0	100	0	0	
Graciosa Island	0	100	0	0		0	100	0	0		0	100	0	0	
Place of Death															
Public heath institution	39.8	42	13.9	4.3	**<0.001**	15.5	42	7.5	35	**<0.001**	4.3	48.7	9.5	37.5	**<0.001**
At home	57	30.8	9.8	2.4		12.2	57.7	4.2	25.9		2.1	63.3	8.7	25.9	
Nursing home	34.6	51.8	11.4	2.1		16.1	46.8	10	27.1		2.5	55	10.7	31.8	
Private health institution	18.2	63.6	18.2	0		27.3	27.3	9.1	36.4		9.1	45.5	9.1	36.4	
Unknown	46.3	46.3	7.3	0		17.1	53.7	4.9	24.4		4.9	56.1	2.4	36.6	
Time Periods															
T1	42.7	40	13.8	3.5	**0.017**	14	47	6.4	32.6	**0.004**	3.5	52.5	9.2	34.8	0.326
T2	34.1	46.6	11.7	7.6		18.6	35.2	5.3	40.9		4.9	46.2	9.8	39	
T3	40	42.3	13.7	4		15.6	42.4	7.8	34.2		4.2	49.2	9.5	37.1	

Notes: Significant results are presented in bold font. Abbreviations: ARS, administrative region of health; AR, autonomous region. Periods: T1—2 March–2 June; T2—3 June–11 August; and T3—12 August–31 December.

## Data Availability

E-death certificates’ Data analyzed is owned by Direção-Geral da Saúde (MoH, Portugal) it is not publicly available but can be requested to geral@dgs.min-saude.pt, the institution has a specific procedure to provide data that needs to be followed and approved by the Director-General of Health. All additional used data are available at the provided sites and links.

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
