# Peer review of "Multimorbidity Profile of COVID-19 Deaths in Portugal during 2020"

_jcm, 2022, doi:10.3390/jcm11071898_

Round 1

Reviewer 1 Report

The authors have done a very thorough job in this study. They have taken on significant amount of analyses with the given dataset and presented a sound discussion section. I particularly appreciate the discussion of Elixhauser vs. Charlson lists, as this was alluded to many times in the results. 

I would recommend trying to edit the English language and style to present this study more succinctly. The same would be recommended for the results - there is a lot of information, much of which is reference in the results section. If there is a way to limit some of the tables to the top 5 comorbidities, or the most common ones (i.e. Table S2), this may be a good way to avoid redundancy of presented results. Overall, this is a very thorough section and supplement. Please note that some of the referenced tables in the body of the text have an error message - I suspect this is due to some technical issue.

Author Response

The authors have done a very thorough job in this study. They have taken on significant amount of analyses with the given dataset and presented a sound discussion section. I particularly appreciate the discussion of Elixhauser vs. Charlson lists, as this was alluded to many times in the results. 

Response: the authors are thankful for the kind words and the appreciation.

I would recommend trying to edit the English language and style to present this study more succinctly. The same would be recommended for the results - there is a lot of information, much of which is reference in the results section. If there is a way to limit some of the tables to the top 5 comorbidities, or the most common ones (i.e. Table S2), this may be a good way to avoid redundancy of presented results. Overall, this is a very thorough section and supplement. Please note that some of the referenced tables in the body of the text have an error message - I suspect this is due to some technical issue.

Response:

  1. in what concerns language the authors just used MDPI editing services to guarantee language improvements. Alterations proposed by the English editing service are kept in manuscript resulting from this revision.
  2. Given Reviewer #1 and #2 requests concerning tables 2 and 3. We opted to reduce table S2 to the first 10 diseases – putting the complete table in Supplement material. Table 3 – concerning Charlson’s diseases was removed from the manuscript completely and just made available in supplemented material Table S5.
  3. Errors appearing across the manuscript were edited and changed to the meant tables or figures references.

Reviewer 2 Report

Congratulations on a very well-written and pertinent, comprehensive article describing the complex morbidity profile of COVID-19 deaths in Portugal. I have two main queries:

(1) I wonder why you present results from both co-morbidity indexes? It may simplify the paper and make your message stronger to just present the Elixhauser but to mention that you had also used the Charleson Index as a supplementary one and that the results were similar. (Unless it was because of dementia? But even so, it would simplify the paper; or leave its details in the supplementary file?)

(2) It is not my specific area of expertise but my understanding is that both of these indexes are used for predicting in-hospital death (rather than any death). It is probably a fair assumption that most deaths from COVID-19 during your study period occurred in patients who had been hospitalised, but if they were not, would this have made a big difference? Do you have any idea what proportion of the deaths you investigated were in hospitalised patients? Could you add a comment on the use of the Elixhauser/Charleson Index in non-hospitalised patients as a potential limitation?

Author Response

Congratulations on a very well-written and pertinent, comprehensive article describing the complex morbidity profile of COVID-19 deaths in Portugal. I have two main queries:

Response: the authors are most appreciated for the kind words.

  • I wonder why you present results from both co-morbidity indexes? It may simplify the paper and make your message stronger to just present the Elixhauser but to mention that you had also used the Charleson Index as a supplementary one and that the results were similar. (Unless it was because of dementia? But even so, it would simplify the paper; or leave its details in the supplementary file?)
  1. Response: Well, at the beginning the Charlson’s commodities had in the manuscript as many tables as Elixhausers’ – a long effort to reduce the extensive results was already in place. Indeed, it was the slight little differences that dragged our will to include some minimal results of the Charlson’s approach. Since both reviewers asked for simplification of concerning these results tables, we have proceeded to some changes accordingly. Namely, we reduce table 2 to the first 10 diseases – putting the complete table in Supplement material. Table 3 – concerning Charlson’s diseases was removed from the manuscript completely and just made available in supplemented material Table S5.

(2) It is not my specific area of expertise but my understanding is that both of these indexes are used for predicting in-hospital death (rather than any death). It is probably a fair assumption that most deaths from COVID-19 during your study period occurred in patients who had been hospitalised, but if they were not, would this have made a big difference? Do you have any idea what proportion of the deaths you investigated were in hospitalised patients? Could you add a comment on the use of the Elixhauser/Charleson Index in non-hospitalised patients as a potential limitation?

Response: Unfortunately, we did not have data to predict death as we had not morbidity information on people who recovered from COVID-19 to compare with deaths. Therefore, we used Elixhauser/Charlson indexes only to describe the morbidity profile of COVID-19 deceased people in Portugal during 2020. Indeed, these comorbidity indexes had already been used to predict outcomes in hospital patients, for instance in hospital mortality, length of hospital stay, or readmissions (e.g., https://pubmed.ncbi.nlm.nih.gov/31094947/). However, we used them to describe morbidity as they were adapted to use with administrative data based on International Classification of Diseases diagnosis codes, as is the case of SICO database. In addition, the most of our sample corresponds to deaths within hospitalized patients (about 91%, please see Location of death in table 1).

To directly reply to the reviewer #2 concern, we added the following sentence in the Discussion. “Both the Elixhauser and Charlson indexes were essentially here used to identify diseases/ comorbidities associated with COVID-19 deaths. As they are usually seen as in-hospital mortality predictive tools, and about 90% of COVID-19 deaths were observed as occurring in a health institution, this might be considered a limitation in our study. Nevertheless, this study was not focused on in-hospital mortality predictions.”

This manuscript is a resubmission of an earlier submission. The following is a list of the peer review reports and author responses from that submission.

Round 1

Reviewer 1 Report

The study under review is a descriptive, cross-sectional study of COVID-19 deaths in Portugal in 2020. This study provides national trends for demographic and comorbidities of patients with associated mortality data. Overall, the manuscript is well written with clear description of results and methods. Statistical analysis section is particularly well laid out. The discussion section is very thorough and includes appropriate rationale for the methodology. The authors have clearly performed thorough analyses. The following points may help improve the manuscript:

  1. On page 2, lines 61-62 (introduction) – consider putting the Portugal numbers into their own sentence as to separate the global introduction from that of the national data.
  2. On page 2, line 67 (introduction) – please clarify the meaning of “mean of deaths” (i.e. is this just the mortality rate in the country, or is it deaths due to a specific cause)
  3. On page 2, lines 76-77 (introduction) – consider rephrasing “exert an important impact on the progression of the clinical status and to the endpoint of the disease” to make this more concise.
  4. Run-on sentences/long sentences have been noted at various points throughout the manuscript (i.e. page 3, lines 125-128) – please try to break these sentences up as you are able.
  5. On page 4, section 3.1 (death characteristics), it is somewhat unclear as to the utility of including both Figures 1A and 1B. If 1B is to address differences in sex, please add labels to Figure 1B to better clarify this.
  6. On page 11, lines 252 (Table 3) – there is a lot of excellent information in this table, however, it is a bit difficult to read given the coding. If it is possible to do so, would recommend consolidating the number of listed conditions to the top 10, or the most important that you have seen in other literature so that the salient points are not lost.
  7. On page 17, lines 369-372 – suggest rewording this as the meaning of this sentence is not initially very clear (I am assuming the meaning of this sentence is to show that Elixhauser is statistically superior to Charlson).

Author Response

Dear Reviewer #1 of Journal of Clinical Medicine

We thank you for the comments made to the manuscript entitled “Sociodemographic and clinical characteristics of 6701 Covid-19 deaths in Portugal during 2020: a descriptive analysis”. We have thoroughly revised the manuscript jcm-1362525, and below we present our responses to your comments. Our responses are denoted in italics.

We thank you for your suggestions, which have undoubtedly improved our manuscript.

Thank you in advance.

Best regards,

Paulo Jorge Nogueira

Universidade de Lisboa, Faculdade de Medicina

Instituto de Medicina Preventiva e Saúde Pública

Avenida Professor Egas Moniz, 1649-028 Lisboa, Portugal

Reviewer #1

Comment: The study under review is a descriptive, cross-sectional study of COVID-19 deaths in Portugal in 2020. This study provides national trends for demographic and comorbidities of patients with associated mortality data. Overall, the manuscript is well written with clear description of results and methods. Statistical analysis section is particularly well laid out. The discussion section is very thorough and includes appropriate rationale for the methodology. The authors have clearly performed thorough analyses.

The authors appreciate these words of reviewer 1 regarding our work. We have reviewed the manuscript and the answers to the reviewer’s comments are provided below.

Comment: On page 2, lines 61-62 (introduction) – consider putting the Portugal numbers into their own sentence as to separate the global introduction from that of the national data.

COVID-19 number of cases and deaths in Portugal were separated and moved to the second paragraph (lines 65-66). Moreover, the overall number of COVID-19 cases and deaths were updated as of 8th September 2021.

Comment: On page 2, line 67 (introduction) – please clarify the meaning of “mean of deaths” (i.e. is this just the mortality rate in the country, or is it deaths due to a specific cause)

The sentence was changed to clarify it. The authors meant that the number of all-cause deaths increased when compared with the average number of all-cause deaths in the previous five years reported in Portugal.

Comment: On page 2, lines 76-77 (introduction) – consider rephrasing “exert an important impact on the progression of the clinical status and to the endpoint of the disease” to make this more concise.

The authors thank this contribution and changed the sentence to make it clearer.

Comment: Run-on sentences/long sentences have been noted at various points throughout the manuscript (i.e. page 3, lines 125-128) – please try to break these sentences up as you are able.

The authors addressed this suggestion of the reviewer and reworded long sentences all over the manuscript: lines (as in the original version of the manuscript) 108-109, 125-130, 184-190, 197-200, 209-215, 240-245, 286-289, 321-326, 334-338, 372-375, 394-398, 398-403, 412-416, 434-438, 441-444, 444-451, and 465-469. Other English edition changes were conducted throughout the text in this version of the manuscript and are marked up using the “Track Changes” function.

Comment: On page 4, section 3.1 (death characteristics), it is somewhat unclear as to the utility of including both Figures 1A and 1B. If 1B is to address differences in sex, please add labels to Figure 1B to better clarify this.

We apologize for the mistake. Figure 1A was incorrectly repeated as figure 1B. The correct figure was already inserted.

Comment: On page 11, lines 252 (Table 3) – there is a lot of excellent information in this table, however, it is a bit difficult to read given the coding. If it is possible to do so, would recommend consolidating the number of listed conditions to the top 10, or the most important that you have seen in other literature so that the salient points are not lost.

Thank you for raising this point. Similar suggestion was proposed by reviewer 2. We changed tables in conciliation of both suggestions.

Comment: On page 17, lines 369-372 – suggest rewording this as the meaning of this sentence is not initially very clear (I am assuming the meaning of this sentence is to show that Elixhauser is statistically superior to Charlson).

The reviewer is right about the meaning of this sentence. The authors reworded it in this manuscript version to make the paragraph clearer.

Additional changes.

The authors moved some paragraphs to better fit tables within pages. The authors also removed some text of the original MDPI template that was wrongly left in the final disclosures of the previous version of the manuscript (e.g., Author Contributions or Appendix).

Reviewer 2 Report

Nogueira et al conducted an interesting cross-sectional study to summarize the sociodemographic and clinical characteristics of people who died of covid19 in Portugal. Here are the comments to further improve the manuscript

Major concern:

1) Extensive English language edit is needed. 

2) Population death rate is an interesting epidemiologic parameter, however, the infection fatality rate is probably more interesting and relevant to health care. 

Minor concern:

1) typo in title "sociodemografic" -> "sociodemographic"

2) Page 2 line 67 99.356 -> 99,356

3) Table 3: full condition name is suggested to be provided in the table rather than in the footnote in order to provide readability. 
